# Digital Twins to Predict Crack Propagation of Sustainable Engineering Materials under Different Loads

Xu Li [1], Gangjun Li [1,*] and Zhuming Bi [2]

1   School of Intelligent Manufacturing, Chengdu Technological University, Chengdu 610031, China; lxu1@cdtu.edu.cn
2   Department of Civil and Mechanical Engineering, Purdue University Fort Wayne, Fort Wayne, IN 46805, USA; biz@pfw.edu
*   Correspondence: lgjun1@cdtu.edu.cn

**Abstract:** *Computer-aided engineering* (CAE) is an essential tool in a digital twin not only to verify and validate a virtual twin before it is transformed into a physical twin, but also to monitor the use of the physical twin for enhanced sustainability. This paper aims to develop a CAE model for a digital twin to predict the fatigue life of materials. Fatigue damage is represented by the size of a macro-crack that grows with a cluster of micro-cracks subjected to three different loads. The growth angle is related to the maximum circumferential tensile stress, and the growth rate is determined by the *stress intensity factor* (SIF) at the crack tip. The prediction model takes into consideration the main factors, including micro-cracks, crack closures, and initial configurations. Simulations are developed for the growth of macro-cracks with *radially distributed micro-cracks* and *randomly distributed micro-cracks*, and we find that (1) the macro-crack in the second case grows faster than that in the first case; (2) a pure shear load affects the macro-crack propagation more than a combined shear and tensile load or a tensional load; (3) the external stresses required to propagate are reduced when the inclination angle of the micro-crack is small and within ($-25° < \beta < 25°$); (4) micro-cracks affect the propagating path of the macro-crack and generally guide the direction of propagation. The developed model has been verified and validated experimentally for its effectiveness in predicting the fracture or fatigue damage of a structure.

**Keywords:** computer-aided engineering (CAE); digital twins; stress intensity factor (SIF); engineering materials; fatigue damage; micro-cracks; crack propagation; verification and validation (V&V)



## 1. Introduction

Pursuing sustainability in design and manufacturing implies that the capability of an engineering material in a product, structure, or system must be fully utilized to reduce or avoid any waste. This cannot be achieved without developing a digital twin for a physical twin so that the physical system (1) is built when its virtual model has been fully verified and validated and (2) is constantly monitored to respond to any conditional changes for maintenance or remedies as soon as possible [1–3]. Since physical products or systems are mainly made from engineering materials, we are especially interested in developing a generic CAE tool as a part of a digital twin to predict the fatigue damage of engineering materials. This can be an integral model of a digital twin where fatigue failure can be a major source of product failure, especially for aerospace products [4].

This paper focuses on fatigue prediction, which is associated with the accumulated damage of crack, and can be measure based on crack size. It is rare that fatigue damage consists of a single crack; in fact, numerous micro-cracks have been found near the tip and the path of a macro-crack using scanning electron microscopy in 7075-T6 aluminum alloy, X65 steel, and U71Mn steel [5–7]. Depending on their relative sizes, cracks are referred to as *macro-cracks* and *micro-cracks*. We are especially interested in the effects of micro-cracks on macro-cracks that are subject to various loading patterns.

Micro-cracks can enhance or reduce the fatigue strength of materials depending on a number of factors. To investigate the effects of fatigue strength on micro-cracks, the complex potential method and pseudo-traction method can be used to model the interactions between macro- and micro- cracks [8–10]. Kachanov [11,12] developed an interaction matrix and represented the effects of micro-cracks on a macro-crack. Tamuzs and Petrova [13,14] used the small parameter method to study the effect of a cluster of cracks on a macro-crack. Li et al. [15–17] assumed that micro-cracks were arbitrarily distributed and adopted the distributed dislocation technique to estimate their impacts on a macro-crack. The *extended finite element method* (XFEM) was used by Loehnert and Belytschko [18] and Sutula et al. [19] to analyze the effects of micro-cracks.

While the afore-mentioned works discussed the interactions between a macro-crack and micro-cracks, none of them considered the relationship and propagation of micro-cracks. We argue *linear elastic fracture mechanics* (LEFM) should be adopted to investigate how micro-cracks are propagated.

The direction and growth rate are two critical factors in propagating cracks [20–23]. Kahn [24] analyzed the impact of the initial crack angle in the mixed crack mode subject to different loads and concluded that the extension of cracks depended on the direction of stress with respect to the crack face. Wang [25] developed analytical and numerical solutions to the stress distribution in the surrounding area of a crack and found that the stress distribution was affected by the loadings and geometrical variations. *Finite element modelling* (FEM) was used by Ghaffari [26] to study the initiation and propagation of cracks in gears. These works ignored the conditions in which micro-cracks pre-existed. In general, a fracture in the real world is mostly caused by the propagation of a single macro-crack and is affected by pre-existing micro-cracks [7,27–30].

Many researchers have contributed to the study of initiation, propagation, and coalescence of cracks. It has been generally agreed upon that crack propagation is closely associated with fatigue fractures. For example, Li et al. [31] applied the distributed dislocation technique to variations in *plastic zones* (PZ) and the locations of micro-cracks. They found that the micro-cracks in front of a propagated macro-crack did not affect the growth of the macro-crack. Basoglu [32], Yates [33], and Liu [34] investigated the changes in a micro-crack when multiple micro-cracks were presented. The experimental methods were also used to investigate the initiation of and changes in macro-cracks [35–38]. There was a need to model the effects of micro-cracks on the fatigue growth of macro-cracks.

The rest of the paper is organized as follows. Section 2 presents a model of an infinite plane that consists of a macro-crack and multiple micro-cracks. The Muskhelishvili's complex potential method is used to find the solution, subject to various loading conditions. Section 3 describes the propagations of cracks, including a macro-crack and multiple micro-cracks. Section 4 verifies the results of the proposed solutions. Section 5 summarizes the effects of micro-cracks on a macro-crack and provides a summary with conclusions.

## 2. Crack Propagation Modelling and Solution

In this section, a generalized representation of an engineering material with accumulated fatigue damage is presented, and the Muskhelishvili's complex potential method is used to find the solution of crack propagation subject to various loading conditions.

### 2.1. Problem Description

To maintain generality, this paper models an engineering material as an elastic plane with one macro-crack and a set of arbitrarily oriented and distributed micro-cracks subjected to a uniaxial tensional stress of $\sigma^\infty$ and a shear stress of $\tau^\infty$, as shown in Figure 1.

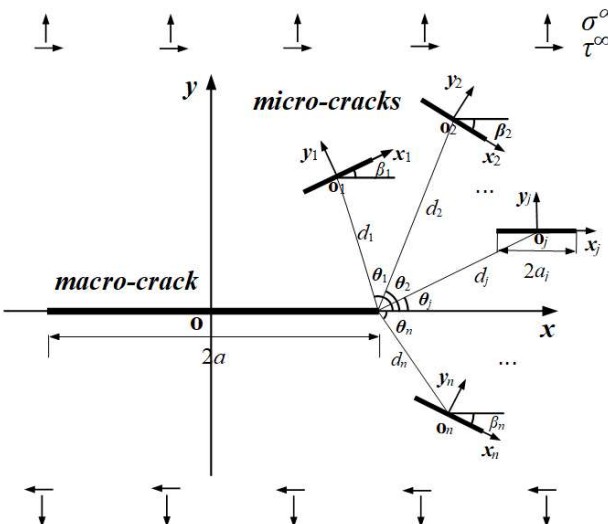

**Figure 1.** A plane with one macro-crack and a set of micro-cracks subject to mixed loads.

To align the propagation of the macro-crack to the set of micro-cracks shown in Figure 1, we decompose the original problem in Figure 1 into three sub-problems, as shown in Figure 2, and look into the corresponding solutions.

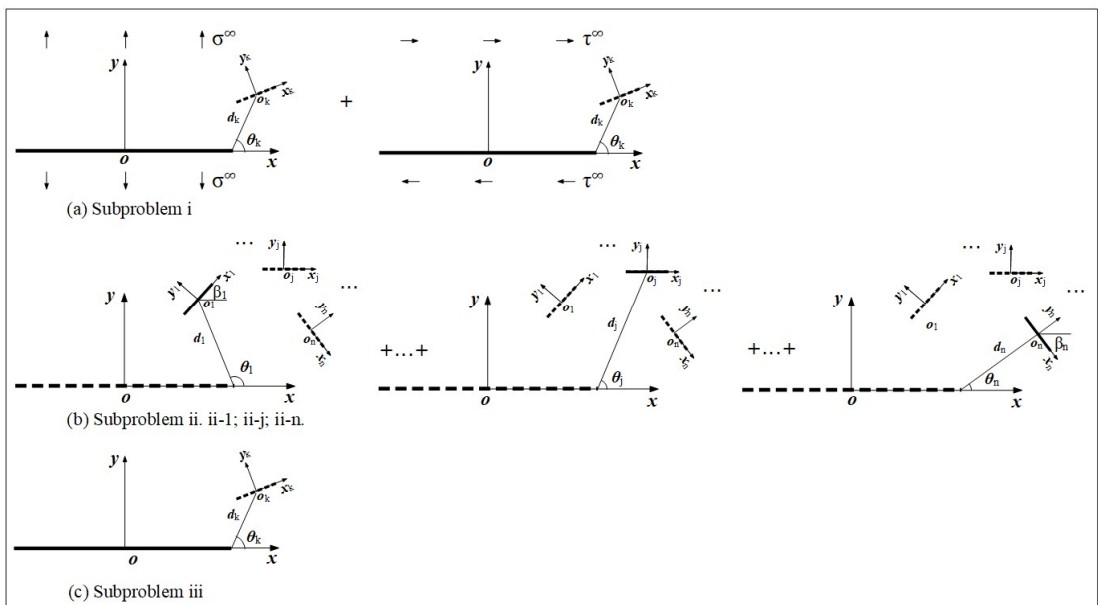

**Figure 2.** (**a**–**c**) Decomposition of the original problem into a set of sub-problems, where the dashed line represents the absence of cracks and the solid line represents the real crack.

Figure 2 shows that the original problem is decomposed into three sub-problems. The macro-crack is sized based on its length ($2a$). The micro-crack $k$ is sized based on its length ($2a_k$), where $k$ is the index of the micro-crack. The global coordinate system ($o$-$x$-$y$) is established with its origin ($o$) at the center of the macro-crack, and $x$ and $y$ are aligned with the length and perpendicular direction of the macro crack, respectively. A local coordinate system ($o$-$x_k$-$y_k$) is established with its origin ($o_k$) at the center of micro-crack $k$, and $x_k$ and $y_k$ are aligned with the length and perpendicular direction of micro-crack $k$, respectively. Moreover, $\beta_k$ and $\theta_k$ denote the inclined orientational angle of micro-crack $k$, respectively, and $d_k$ is the distance from $o$ to $o_k$.

The Muskhelishvili's complex potential method is adopted to solve the formulated problems, and the solutions must satisfy all the boundary conditions applied to the cracks.

For *sub-problem* (i), the stress distribution surrounding micro-crack *j* is determined based on the stress function available in ref. [30]. For *sub-problem* (ii), the resulting stresses apply to micro-crack *j* in the opposite direction. Due to the coupling of multiple micro-cracks, the stress on micro-crack *j* exists, but with a small magnitude. For *sub-problem* (iii), the solution includes both the redundant stress of micro-crack *j* from sub-problem (ii) and that at the position of non-existent micro-crack *j*. For *sub-problem* (iv), the stress of non-existent micro-crack *j* obtained from the solution to sub-problem (iii) is applied to real micro-crack *j*.

### 2.2. Solving Procedure

Assume the material is isotropic and elastic and its deformation on the plane of interest is within the elastic range of the material, the stress distribution around a crack is expressed by two complex potentials, $\phi(z)$ and $\psi(z)$ [39], as follows:

$$\left.\begin{array}{l} \sigma_{xx} + \sigma_{yy} = 2\left\{\phi'(z) + \overline{\phi}'(z)\right\} \\ \sigma_{xx} - \sigma_{yy} + 2i\tau_{xy} = 2\{\overline{z}\phi''(z) + \psi'(z)\} \end{array}\right\} \tag{1}$$

where $z = x + iy$; $\sigma_{xx}$ and $\sigma_{yy}$ are normal components along the horizontal and vertical directions, respectively; $\tau_{xy}$ is the shear stress; and $(')$ and $('')$ are the operations for the first and second derivatives with respective to $z$, respectively.

#### 2.2.1. Solution to Sub-Problem (i) for the Macro-Crack

The stress is distributed as follows [39]:

$$\phi(z) = \frac{i\tau^{\propto}}{2}\left(z - \sqrt{z^2 - a^2}\right), \ \psi(z) = \frac{i\tau^{\propto}a^2}{2}\left(z - \frac{a^2}{\sqrt{z^2 - a^2}}\right) \tag{2}$$

substituting Equation (2) into Equation (1) to find the stress around the tip of a macro-crack. For micro-crack *k*, the stress components $\sigma_{xx(k)}^{(i)}$, $\sigma_{yy(k)}^{(i)}$, and $\tau_{xy(k)}^{(i)}$ can be obtained as follows:

$$\sigma_{xx(k)}^{(i)} = \sigma_{xx}^{\sigma^{\propto}} + \sigma_{xx}^{\tau^{\propto}}, \ \sigma_{yy(k)}^{(i)} = \sigma_{yy}^{\sigma^{\propto}} + \sigma_{yy}^{\tau^{\propto}}, \ \tau_{xy(k)}^{(i)} = \tau_{xy}^{\sigma^{\propto}} + \tau_{xy}^{\tau^{\propto}} \tag{3}$$

Further conducing coordinate transformation, the normal and tangential components of the stress around micro-crack *k* in the local coordinate system are as follows:

$$\left.\begin{array}{l} \sigma_{N(k)}^{MI(i)} = \frac{\sigma_{xx(k)}^{(i)} + \sigma_{yy(k)}^{(i)}}{2} + \frac{\sigma_{xx(k)}^{(i)} - \sigma_{yy(k)}^{(i)}}{2}cos2\beta_k - \tau_{xy(k)}^{(i)}sin2\beta_k \\ \tau_{T(k)}^{MI(i)} = \frac{\sigma_{xx(k)}^{(i)} - \sigma_{yy(k)}^{(i)}}{2}sin2\beta_k + \tau_{xy(k)}^{(i)}cos2\beta_k \end{array}\right\} \ (k = 1, 2, \ldots n) \tag{4}$$

where subscripts *N* and *T* denote the normal and tangential direction of the crack, respectively.

#### 2.2.2. Solution to Sub-Problem (ii) for Micro-Cracks

We find the sub-solution one micro-crack at a time; if there are *k* micro-cracks, the solving procedure will be repeated *k* times. In each step, it is assumed that an infinite plane contains a single micro-crack *j*, one virtual macro-crack, and *k* − 1 virtual micro-cracks. It is further assumed that micro-crack *j* is subjected to a non-uniform distributed load of $\sigma_{N(j)}^{MA(i)}$ and $\tau_{T(j)}^{MA(i)}$ from the solution in Section 2.2.1, but with the opposite direction.

Firstly, the load is treated as an assembly of numerous small, concentrated forces *F*, and the stress due to *F* applied to the upper and lower boundary of the crack becomes [39]:

$$\phi(\zeta) = \frac{F}{2\pi i}\left\{\frac{\zeta_0}{\zeta(\zeta_0^2 - 1)} - \frac{\kappa}{\kappa + 1}ln\zeta - \ln(\zeta_0 - \zeta) + \frac{\zeta_0^2\left(a\left(\zeta + \frac{1}{\zeta}\right) - 2z_1\right)}{a(\zeta_0 - \zeta)(\zeta_0^2 - 1)}\right\}\frac{i(\zeta_0^2 - 1)}{\zeta_0|\zeta_0 - \overline{\zeta}_0|}$$

$$\psi(\zeta) = \frac{F}{2\pi i}\left\{\frac{2\zeta\zeta_0 - \zeta_0^2(\zeta^2 + 1)}{(\zeta^2 - 1)(\zeta_0^2 - 1)} + \frac{ln\zeta}{\kappa + 1} + \frac{2}{(\kappa + 1)(\zeta^2 - 1)} - \ln(\zeta_0 - \zeta) - \frac{2\overline{z}_1\zeta_0^2}{a(\zeta_0 - \zeta)(\zeta_0^2 - 1)}\right\}\frac{i(\zeta_0^2 - 1)}{\zeta_0|\zeta_0 - \overline{\zeta}_0|} \tag{5}$$

The concentrated forces $F$ in the local coordinate system $O_j x_j y_j$ correspond to the distributed load [39]:

$$F = P + iQ = \left[ -\sigma_{N(j)}^{MI(i)} dx_F \right] + i \left[ -\tau_{T(j)}^{MI(i)} dx_F \right] \tag{6}$$

where $P$ and $Q$ are the components in the normal and tangential directions, respectively. Equation (6) can be rewritten using the conformal mapping method to transform $\phi(z)$ and $\psi(z)$.

The stress components $\sigma_{xxF}^{MA(ii)}$, $\sigma_{yyF}^{MA(ii)}$, $\tau_{xyF}^{MA(ii)}$, $\sigma_{xxF}^{MI(ii)}$, $\sigma_{yyF}^{MI(ii)}$, and $\tau_{xyF}^{MI(ii)}$ acting on the virtual macro-crack and micro-cracks due to F can be obtained using Equations (1), (4), and (5). The coordination transformation can be applied to find these stress components in the local coordinate system $O_j x_j y_j$ as follows:

$$\left. \begin{array}{c} \sigma_{xx(j)}^{MA(ii)}, \sigma_{yy(j)}^{MA(ii)}, \tau_{xy(j)}^{MA(ii)} = \int_{-a_k}^{a_k} \left( \sigma_{xxF}^{MA(ii)}, \sigma_{yyF}^{MA(ii)}, \tau_{xyF}^{MA(ii)} \right) dx_F \\[2mm] \sigma_{xx(k)}^{MI(ii)}, \sigma_{yy(k)}^{MI(ii)}, \tau_{xy(k)}^{MI(ii)} = \int_{-a_k}^{a_k} \left( \sigma_{xxF}^{MI(ii)}, \sigma_{yyF}^{MI(ii)}, \tau_{xyF}^{MI(ii)} \right) dx_F \\[2mm] \sigma_{xx(k)}^{MI(ii)}, \sigma_{yy(k)}^{MI(ii)}, \tau_{xy(k)}^{MI(ii)} = 0 \quad (k = j) \end{array} \right\} (k = 1, 2, \ldots n, k \neq j) \tag{7}$$

For the macro-crack, subscript $(j)$ shows the contribution of micro-crack $j$ to the macro-crack. For a micro-crack, subscript $(k)$ shows the contribution of micro-crack $j$ to the $k - 1$ virtual micro-crack.

Next, transforming the results from $O_j x_j y_j$ to the global coordinate system o-x-y and local coordinate systems for a micro-crack as $O_k x_k y_k (k, 1, 2, \ldots, n, )$, $\sigma_{N(j)}^{MI(ii)}, \tau_{T(j)}^{MI(ii)}, \sigma_{N(k)}^{MI(ii)}$, and $\sigma_{T(k)}^{MI(ii)}$ can be obtained when the stress is applied to micro-crack $j$. The same procedure applies to any of other $k - 1$ micro-cracks as follows:

$$\left. \begin{array}{c} \sigma_N^{MA(ii)} = \sum_{j=1}^{n} \sigma_{N(j)}^{MA(ii)} \\[2mm] \tau_T^{MA(ii)} = \sum_{j=1}^{n} \sigma_{T(j)}^{MA(ii)} \end{array} \right\} \tag{8}$$

$$\left. \begin{array}{c} \sigma_N^{MI(ii)} = \sum_{k=1}^{n} \sigma_{N(k)}^{MI(ii)} \\[2mm] \tau_T^{MI(ii)} = \sum_{k=1}^{n} \sigma_{T(k)}^{MI(ii)} \end{array} \right\} (k = 1, 2, \ldots n) \tag{9}$$

Taking into consideration of effects of other micro-cracks, the stress in Equation (8) is not zero. However, the stress magnitude in each micro-crack is insignificant in comparison with $\sigma_{N(j)}^{MI(i)}$ and $\tau_{T(j)}^{MI(i)}$, which were solution to sub-problem (i).

### 2.2.3. Solution to Sub-Problem (iii) for the Macro-Crack

Here, the effects of one macro-crack in the plane are investigated. The macro-crack is subjected to a non-uniformly distributed stress whose magnitude and direction are determined in Equation (7). The sample procedure in Section 2.2.2 is applied. When a concentrated force $F$ is applied to the macro-crack, $\sigma_{xxF}^{MI(iii)}, \sigma_{yyF}^{MI(iii)}$, and $\tau_{xyF}^{MI(iii)}$ on a virtual micro-crack are obtained as follows [39]:

$$F = P + iQ = \left[ -\sigma_N^{MA(ii)} dx_F \right] + i \left[ -\tau_T^{MA(ii)} dx_F \right] \tag{10}$$

The stress components in micro-crack $k$ are as follows:

$$\sigma_{xx(k)}^{MI(iii)}, \sigma_{yy(k)}^{MI(iii)}, \tau_{xy(k)}^{MI(iii)} = \int_{-a_k}^{a_k} \left( \sigma_{xxF}^{MI(iii)}, \sigma_{yyF}^{MI(iii)}, \tau_{xyF}^{MI(iii)} \right) dx_F \quad (k = 1, 2, \ldots n) \tag{11}$$

The normal and shear stresses $\sigma_{N(k)}^{MI(iii)}$ and $\tau_{T(k)}^{MI(iii)}$ in $O_k x_k y_k$ $(k, 1, 2, \ldots, n,\ )$ can be obtained through coordinate transformation, and the stresses of the virtual micro-cracks are obtained in this step.

### 2.2.4. Solution to Sub-Problem (iv) from Micro-Cracks

The stresses of micro-cracks have been found based on the solutions to sub-problems (ii) and (iii); therefore, the loads by the micro-cracks become $-\left(\sigma_{N(k)}^{MI(ii)} + \sigma_{N(k)}^{MI(iii)}\right)$ and $-\left(\tau_{T(k)}^{MI(ii)} + \tau_{T(k)}^{MI(iii)}\right)$, and, similar to Equations (7) and (8), the stress components of $\sigma_N^{MA(iv)}$, $\tau_T^{MA(iv)}$, $\sigma_{N(k)}^{MI(iv)}$ and $\tau_{T(k)}^{MI(iv)}$ $(k = 1, 2, \ldots, n)$ can be obtained.

The aforementioned procedure shows that solving sub-problem (i) leads to the stress components of $\sigma_{N(k)}^{MI(i)}$ and $\tau_{T(k)}^{MI(i)}$. Solving the sub-problem leads to the stress components of $-\sigma_{N(k)}^{MI(i)}$ and $-\tau_{T(k)}^{MI(i)}$ on micro-cracks; moreover, $\sigma_{N(k)}^{MI(ii)}$ and $\tau_{T(k)}^{MI(ii)}$ in Equation (8) are negligeable in comparison with $\sigma_{N(j)}^{MI(i)}$ and $\tau_{T(j)}^{MI(i)}$ from the solution to sub-problem (i). Solving sub-problem (iii) leads to the stress components of $\sigma_{N(k)}^{MI(iii)}$ and $\tau_{T(k)}^{MI(iii)}$.

To define the constraints in sub-problem (iv), $-\sigma_{N(k)}^{MI(ii)}$, $-\tau_{T(k)}^{MI(ii)}$, $-\sigma_{N(k)}^{MI(iii)}$, and $-\tau_{T(k)}^{MI(iii)}$ are summed as the stresses due to micro-cracks. Note that the stress components $\sigma_{N(k)}^{MI(iv)}$ and $\tau_{T(k)}^{MI(iv)}$ are negligible in comparison with $\sigma_{N(k)}^{MI(iii)}$ and $\tau_{T(k)}^{MI(iii)}$. The sub-problems can be further decomposed to improve the accuracy of the developed models.

In the proposed model, the coupling among the cracks is considered for all sub-problems. Sub-problem (i) has modelled the effects of the macro-crack on the micro-cracks; sub-problem (ii) has modelled the effects of the micro-cracks on the macro-crack; sub-problem (iii) has re-refined the effects of the macro-crack on the micro-cracks, and this procedure can be iteratively performed until the accuracy becomes satisfactory.

### 2.3. Solution to $K_I$ and $K_{II}$

Traction-free boundary conditions on the crack surface can be satisfied using a step-by-step procedure of problem decomposition, as discussed in Section 2.2, since the solution to each sub-problem can be obtained using LEFM independently.

#### 2.3.1. Solution of LEFM

Let the loads on the macro-crack surface be $\sigma_N^{MA(i)} = -\sigma^\propto$ and $\tau_N^{MA(i)} = -\tau^\propto$ in sub-problem (i), and the SIFs at the tip of crack are as follows [39]:

$$\left.\begin{aligned} K_I^{MA(i)} &= \int_{-a}^{a} \frac{-\sigma^\propto}{\sqrt{\pi a}} \left(\frac{a+x}{a-x}\right)^{1/2} dx = \sigma^\propto \sqrt{\pi a} \\ K_{II}^{MA(i)} &= \int_{-a}^{a} \frac{-\tau^\propto}{\sqrt{\pi a}} \left(\frac{a+x}{a-x}\right)^{1/2} dx = \tau^\propto \sqrt{\pi a} \end{aligned}\right\} \tag{12}$$

Using the solutions to sub-problems (i), (iii), and (v) in the previous section, the SIFs at the tip can be further determined as follows:

$$\left.\begin{aligned} K_{I,elastic}^{MA} &= \sigma^\propto \sqrt{\pi a} + \int_{-a}^{a} \frac{-\left[\sigma_N^{MA(ii)} + \sigma_N^{MA(iv)} + \sigma_N^{MA(vi)} \ldots\right] \cdot}{\sqrt{\pi a}} \left(\frac{a+x}{a-x}\right)^{1/2} dx \\ K_{II,elastic}^{MA} &= \tau^\propto \sqrt{\pi a} + \int_{-a}^{a} \frac{-\left[\tau_T^{MA(ii)} + \tau_T^{MA(iv)} + \tau_T^{MA(vi)} \ldots\right] \cdot}{\sqrt{\pi a}} \left(\frac{a+x}{a-x}\right)^{1/2} dx \end{aligned}\right\} \tag{13}$$

The stress components on micro-crack $k$ include (1) $-\sigma_{N(k)}^{MI(i)}$ and $-\tau_{T(k)}^{MI(i)}$ from the solution to sub-problem (ii), (2) $-\left(\sigma_{N(k)}^{MI(ii)} + \sigma_{N(k)}^{MI(iii)}\right)$ and $-\left(\tau_{T(k)}^{MI(ii)} + \tau_{T(k)}^{MI(iii)}\right)$ from the

solution to sub-problem (iv), and (3) $-\left(\sigma_{N(k)}^{MI(iv)} + \sigma_{N(k)}^{MI(v)}\right)$ and $-\left(\tau_{T(k)}^{MI(iv)} + \tau_{T(k)}^{MI(v)}\right)$ from the solution to sub-problem (vi). Therefore, the SIFs at the tip of micro-crack $k$ become:

$$\left.\begin{array}{l} K_{I(k),elastic}^{MI} = \int_{-a_k}^{a_k} \dfrac{-\left[\sigma_{N(k)}^{MI(i)} + \sigma_{N(k)}^{MI(ii)} + \sigma_{N(k)}^{MI(iii)} + \sigma_{N(k)}^{MI(iv)} \cdots\right]}{\sqrt{\pi a_k}} \cdot \left(\dfrac{a_k+x}{a_k-x}\right)^{1/2} dx \\[4mm] K_{II(k),elastic}^{MI} = \int_{-a_k}^{a_k} \dfrac{-\left[\tau_{T(k)}^{MI(i)} + \tau_{T(k)}^{MI(ii)} + \tau_{T(k)}^{MI(iii)} + \tau_{T(k)}^{MI(iv)} \cdots\right]}{\sqrt{\pi a_k}} \cdot \left(\dfrac{a_k+x}{a_k-x}\right)^{1/2} dx \end{array}\right\} \quad (k = 1, 2, \ldots n) \quad (14)$$

### 2.3.2. Correction of SIF for Ductile Materials

Crack propagation will be affected when a yield occurs to a ductile material. Here, the effect of plastic strain is modelled as a correction to SIF by a *plastic zone* (PZ). It is assumed that the plastic strain is relatively small (a >> r) in comparison with the size of the visible crack [4].

*Firstly*, for a crack with a mixed mode, the stress components in three directions are determined for failure mode I and mode II as follows [39]:

$$\sigma_x = \frac{K_I}{\sqrt{2\pi r_1}}, \sigma_y = \frac{K_I}{\sqrt{2\pi r_1}}, \tau_{xy} = \frac{K_{II}}{\sqrt{2\pi r_1}}, \sigma_z = \tau_{xz} = \tau_{zy} = 0 \quad (15)$$

where $K_I$ and $K_{II}$ are evaluated in Equations (12) and (13) for macro- and micro cracks, respectively. Accordingly, the von Mises stress can be found as follows [39]:

$$\sigma_{eq} = \left\{\frac{(\sigma_x - \sigma_y)^2 + (\sigma_x - \sigma_z)^2 + (\sigma_z - \sigma_y)^2 + 6\left(\tau_{xy}^2 + \tau_{yz}^2 + \tau_{xz}^2\right)}{2}\right\}^{1/2} \quad (16)$$

where $\sigma_{eq}$ is the equivalent von Mises stress. The characteristic size $r_1$ of the plastic zone meets the condition of ref. [39]:

$$\sigma_{eq} = \sqrt{\frac{K_I^2 + 3K_{II}^2}{\pi r_1}} = \sigma_s \quad (17)$$

When the material is ideally elastoplastic, $r_1 = r_y$ is small and is added as the extended length of a crack (see Figure 3) as follows [39]:

$$\sigma_{eff} = a + r_y \quad (18)$$

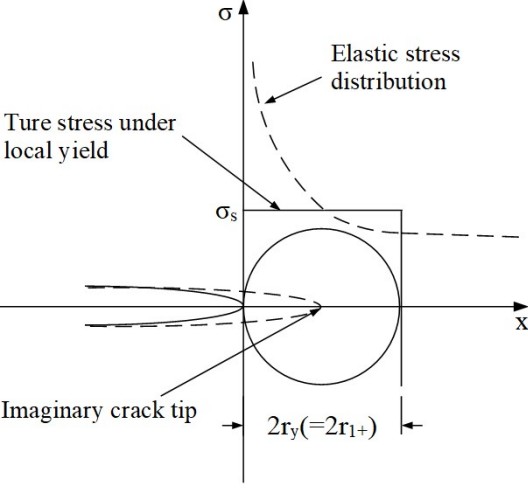

**Figure 3.** The PZ of the Irwin model for mode I.

Figure 3 shows that $r_y$ is doubled due to the effects of stress relaxation. The distance from the crack tip to the tip of an imaginary crack is as follows [39]:

$$r_y = \frac{K_I^2 + 3K_{II}^2}{\pi \sigma_s^2} \tag{19}$$

The resultant stress at the position of the extended length $r_y$ is equal to the yielding strength and is distributed uniformly in Figure 3. Therefore, the normal and tangential stresses of the crack become:

$$\left. \begin{array}{l} \sigma_{n,plas} = \left( \frac{K_I^2}{K_I^2 + 3K_{II}^2} \right)^{1/2} \sigma_s \\[3mm] \tau_{n,plas} = \left( \frac{3K_{II}^2}{K_I^2 + 3K_{II}^2} \right)^{1/2} \sigma_s \end{array} \right\} \tag{20}$$

With the consideration of PZ, the SIFs are corrected as follows:

$$\left. \begin{array}{l} K_{I,plastic} = \int_{-(a+r_{1-})}^{-a} \frac{\sigma_{n,plas}}{\sqrt{\pi a_c}} \left( \frac{a_c+x}{a_c-x} \right)^{\frac{1}{2}} dx + \int_{a}^{a+r_{1+}} \frac{\sigma_{n,plas}}{\sqrt{\pi a_c}} \left( \frac{a_c+x}{a_c-x} \right)^{\frac{1}{2}} dx \\[3mm] K_{II,plastic} = \int_{-(a+r_{1-})}^{-a} \frac{\tau_{n,plas}}{\sqrt{\pi a_c}} \left( \frac{a_c+x}{a_c-x} \right)^{\frac{1}{2}} dx + \int_{a}^{a+r_{1+}} \frac{\tau_{n,plas}}{\sqrt{\pi a_c}} \left( \frac{a_c+x}{a_c-x} \right)^{\frac{1}{2}} dx \end{array} \right\} \tag{21}$$

where $a_c = a + r_{1-} + r_{1+}$ is the modified length of the macro-crack.

## 3. Discussion on Crack Propagation

Fatigue damage is processed at three phases, i.e., *initiation*, *steady propagation* (SP), and *rapid propagation* (RP) [4]. When the size of a crack is less than a threshold, it experiences SP; otherwise, it experiences RP, which rapidly leads to a fracture. It is worth looking into how a crack is propagated in SP and RP.

### 3.1. Steady Propagation (SP)

Based on their size, cracks can generally can be divided into three categories: microstructural short cracks, physical short cracks, and macroscopic cracks. A short crack becomes a macroscopic crack and SP is transformed into RP when the size of a crack reaches a critical length. Crack propagation is affected by many factors such as mean and fluctuated stresses, temperature, and microstructures. Pook [40] and Milella [41] commented that a critical length for metals is usually in a range of (0.25, 0.3) mm. Miller [42] and Stephens et al. [43] indicated that a critical size related to grain size is typically a depth of three to five times the grain size.

Here, we assume that the length of a macro-crack is fixed and above the critical length of a crack while the sizes of micro-cracks are randomly selected and below the critical length of a crack. The threshold SIF of the micro-crack is found using the El Haddad model [44] as follows:

$$\Delta K_{th} = \sqrt{\frac{a_k}{a_k + a_0}} \tag{22}$$

where $a_k$ is the micro-crack length, $\Delta K_{th}$ is the threshold SIF of the micro-crack, and $a_0$ is the critical length of the material, determined by [44]:

$$a_0 = \frac{1}{\pi} \left( \frac{\Delta K_{th}}{\Delta \sigma_o} \right)^2 \tag{23}$$

where $\Delta \sigma_0$ is the fully reversed fatigue strength. Taking an example, Table 1 shows the material properties of U71Mn steel.

**Table 1.** Material properties of U71Mn steel [45].

| Fatigue Limits $\Delta\sigma_0$ | $C$ | $m$ | $\Delta K_{th}$ | $D$ | Yield Limits $\sigma_s$ | $a_0$ |
|---|---|---|---|---|---|---|
| 96 MPa | $4.597 \times 10^{-13}$ | 2.88 | 2.2 MPa·m$^{1/2}$ | 100 μm | 550 MPa | 167 μm |

*3.2. Rapid Propagation (RP)*

The level of a crack closure ($U$) is characterized by:

$$U = \frac{1 - K_{op}/K_{I,max}}{1 - R} \tag{24}$$

where $R$ is the stress ratio, $K_{op}$ is the SIF corresponding to an opened crack, and $K_{I,max}$ is the intensity factor of the mode-I crack.

It is known that the residual PZ at the tip of a crack in the last loading cycle causes a crack closure. Wolf [46] provided an empirical equation to calculate the U of an aluminum alloy as $U = 0.55 + 0.33R + 0.12R^2$. Note that Equation (24) takes into consideration the plastic-hardening effect and $K_{op} = 0.45K_{I,max}$ when $U = 0.55$ and $R = 0$. Therefore, the crack was closure free, and the closure level was $U = 1$.

For RP, the closure level was affected significantly by the stress ratio and propagation modes. Moreover, we took into consideration the impact of mode I and mode II on the SIF in Equation (25) [47] as follows:

$$\Delta K = \sqrt{\Delta K_I^2 + 2\Delta K_{II}^2} \tag{25}$$

*3.3. Propagation Direction*

Here, the fracture was justified based on the maximum circumferential tensile stress, and it was used to predict the direction of propagation [48] as follows:

$$\gamma = -\arccos\left(\frac{3K_{II}^2 + K_I\sqrt{K_I^2 + 8K_{II}^2}}{K_I^2 + 9K_{II}^2}\right) \tag{26}$$

where $\gamma$ is the direction of crack propagation.

For radially distributed micro-cracks, the propagation angle of macro-crack $\gamma$ was $-1.89°$, $-70.26°$, and $-38.86°$ under pure tension, pure shear, and tension-shear loads, respectively. For randomly distributed micro-cracks, $\gamma$ was $5.65°$, $-70.53°$, and $-45.68°$ under pure tension, pure shear, and tension-shear loads, respectively. When the micro-cracks were virtual around the tip of the macro-crack, $\gamma$ was $0°$, $(-70°, -100°)$ and $-40°$ [24], respectively. Thus, the propagation angle was unique to a specified loading condition and damage configuration, and the micro-cracks directly affected the propagation path of the macro-crack. This phenomenon was attributed to the attraction effects of the micro-cracks, inclusion, and dislocations on crack propagation [49,50]. In other words, the micro-cracks guided the propagation of the macro-crack, ultimately leading to a fracture. The macro-crack would coalesce with a favorable micro-crack near the tip of macro-crack to reform a new macro-crack.

**4. Experiment**

The experiment was conducted in this section to verify the results of the developed models based on observation, shown in Figures 4 and 5. The specimen and loading conditions are shown in Figure 4, and U71Mn steel was selected as the material. The specimen was rectangular and was 120 mm in length, 15 mm in width, and 1 mm in height. The macro crack was designed on the edge. It was located in the middle of an edge of the specimen, and its initial length was 2.846 mm. A fluctuating load with a constant amplitude was applied through three-point bending at a loading frequency of 1 Hz. The maximum applied load was 1000 N and the stress ratio was 0.1, recorded using a digital force gauge.

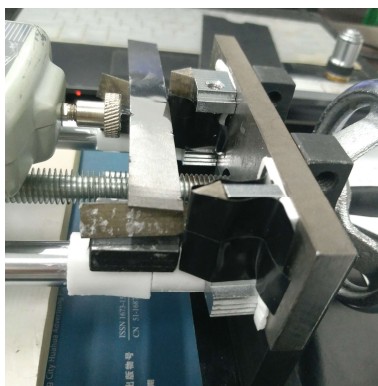

**Figure 4.** U71Mn steel under a three-point bending load.

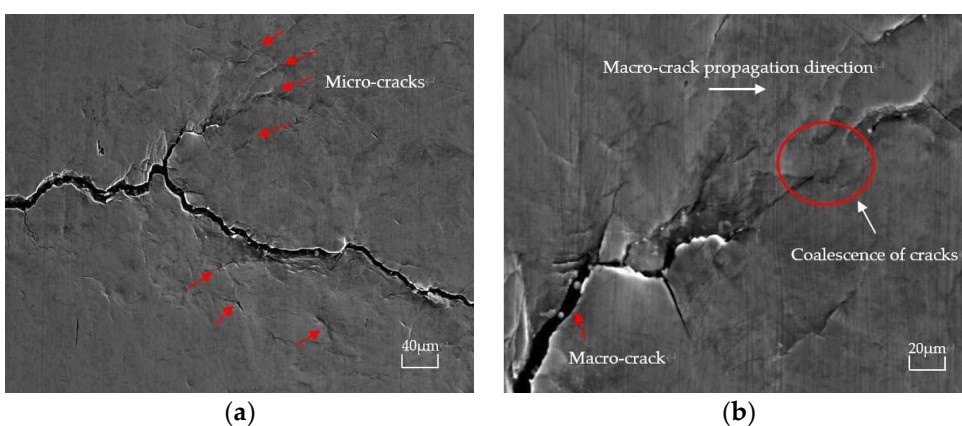

|  (**a**)  |  (**b**)  |
|---|---|

**Figure 5.** Scanning electron microscope photograph of crack propagation. (**a**) Numerous micro-cracks near the macro-crack path; (**b**) coalescence of the macro-crack and micro-cracks.

There were numerous micro-cracks near the propagation path of the macro-crack, as shown in Figure 5a. The micro-cracks on the plane are highlighted with red arrows, and the coalescence of the macro-crack and micro-cracks are highlighted with a red circle. It was observed that the macro-crack was deflected and propagated in a zigzag manner; this was validated by the observations in ref. [27]. The zigzag path of propagation was greatly affected by the presence and propagation of micro-cracks; the micro-cracks guided the propagation of the macro-crack, since it would coalesce with a favorable micro-crack to form an extended macro-crack. According to Section 3.3, we found that (1) for radially distributed micro-cracks, the influence of the micro-cracks on macro-crack propagation was relatively small, the changes in the propagation angle of the macro-crack were small; (2) for randomly distributed micro-cracks, the influence of the micro-cracks on macro-crack propagation was relatively large. The macro-crack path would deflect and coalescence with the micro-cracks that were in the propagation direction.

## 5. Summary and Conclusions

This paper aimed to develop a CAE model for a digital twin to predict the fatigue life of selected engineering materials. Fatigue failure is a major concern in applications involving metal products [4]. In this paper, the solutions of crack propagation for a plane that contained a macro-crack and a number of micro-cracks under various loading conditions were obtained based on highly efficient theoretical models. The effects of micro-cracks on macro-crack growth were analyzed, and the following findings were obtained:

(1) The propagation of a macro-crack and micro-cracks depended mainly on the failure configurations of the micro-cracks in front of the macro-crack as well as the loading conditions.

(2) Micro-cracks with a small inclination angle ($-25° < \beta < 25°$) required less external stress to activate propagation.

(3) Randomly distributed micro-cracks affected the propagation of the micro-cracks and the macro-crack more significantly than radially distributed micro-cracks.

(4) Micro-cracks affected the propagation of the macro-crack primarily due to the pure shear load, followed by the tension-shear load and the pure tension load.

(5) The presence of micro-cracks guided the propagation path of the macro-crack. The macro-crack would coalesce with a favorable micro-crack to form an extended macro-crack.

The influence of micro-cracks on macro-crack propagation is an important subject in fracture mechanics; however, due to the complex calculations of the mutual superposition of micro-cracks, researchers may choose to ignore or approximate the relationship among micro-cracks to predigest the calculation. This approach may create a bottleneck when the number of micro-cracks increases significantly, which can cause an inaccurate result. In the future, we will continue studying the prediction of micro-cracks on macro-crack paths, considering a large number of micro-cracks near the macro-crack tip based on the developed modal, neural network models, or a machine learning method. We also plan to investigate the impacts of micro-crack parameters (size, initial angle, distance between cracks, etc.) on propagation behavior. We expect that the results will be utilized to optimize anti-fracture designs by predicting the fracture behaviors of engineering materials.

**Author Contributions:** Conceptualization, X.L. and G.L.; methodology, X.L. and G.L.; software, X.L.; validation, X.L., G.L. and Z.B.; formal analysis, X.L.; resources, G.L.; writing—original draft preparation, X.L. and Z.B.; writing—review and editing, X.L. and Z.B.; supervision, G.L.; project administration, G.L.; funding acquisition, X.L. and G.L. All authors have read and agreed to the published version of the manuscript.

**Funding:** This work is supported by the Talent Introduction Project of Chengdu Technological University (2022RC005).

**Data Availability Statement:** Data are contained within the article.

**Conflicts of Interest:** The authors declare no conflicts of interest.

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
