# Peer review of "Digital Twins to Predict Crack Propagation of Sustainable Engineering Materials under Different Loads"

_machines, doi:10.3390/machines12020125_

Round 1
Reviewer 1 Report
Comments and Suggestions for Authors
‘Digital Twins to Predict Crack Propagation of Engineering Materials under Different Loads for Sustainable Products’
manuscript Machines (MDPI): machines-2840157-peer-review-v1
by Xu Li, et al.
The title should precisely reflect the content of the paper; the present title is confusing:
1 - The focus of the paper is to solve a fracture mechanics problem: use of LEFM to model FCG of a large crack at the tip of which there are several small cracks with different orientations. Of course fatigue is related to sustainability, but the use of the word sustainability in the title of this paper misleads the reader since it does not reflect the main paper contents.
2 - Equal concern regarding reference to digital twins. The ‘digital twin’ paradigm involves the interplay of numerical / theoretical models and real structures throughout their life-cycle. The present paper concentrates on modelling a fracture mechanics problem, i.e. it may constitute one ‘link’ of the ‘digital twin’; the work is related to digital twins philosophy, but that is certainly not the main emphasis of the paper.
A problem to the paper is the poor quality of use of technical English. This is particularly a problem in the abstract and section 1. They need a full re-writing. The other sections are also full of presentation problems, such as occasional use of words that do not make sense as ‘radically’.
The paper revisits some classical LEFM concepts and tools, as Elasticity based on complex variable following Muskhelishvili and/or Fracture Mechs pioneers as Westergaard. The manuscript includes a large amount of dense equations. The reader would like to know what equations were derived by the present authors, and what other equations are transcribed from the literature. In the last case, the reader wants to have references. These requirements should be taken into account by the present authors in a revision of their manuscript.
References are not presented with a consistent format, and this must be revised/corrected by the authors. More importantly, some references are simply wrong – e.g. refs. 8, 21, whose authors are not properly given/written.
Details:
Examples of main crack with several smaller cracks in the tip region should be provided in the introduction, circa lines 41-42.
Lines 56-57: the use of LEFM with very short cracks is subject to controversy (e.g. concerning length of crack versus plastic region size, grain size, etc.). In lines 56-57 some discussion concerning that controversy, including relevant refs., should be presented.
Lines 67-68: please provide examples and suitable references.
Line 114: case (iv) – why is this case not represented in Figure 2?
Lines 216-217: The authors should not say that it is reasonable to assume that the plastic strain is relatively small. This assumption is implicit in LEFM; therefore they should omit ‘it is reasonable’ and simply state that it is assumed that the plastic region is relatively small.
Typos, etc.:
|
line |
where it is |
should be |
|
13 |
... predict a fatigue life ... |
... predict fatigue life... |
|
13 |
... since any product and structure is made of selected engineering materials. ... |
omit this sentence; it is obvious and does not add anything to the text. |
|
15 |
... is corresponded to the maximum .... |
... is related to the maximum .... |
|
17 |
... initialized angle ... |
.... initial angle ... |
|
18 |
... radically ... |
please correct: ‘radically’ does not make any sense! |
|
41 |
.... damage occurs to single crack ... |
... damage consists of a single crack .... |
|
60 |
Shafique [20] |
Kahn [20] |
|
62 |
Lwa [21] |
Wang [21] |
|
62 |
.... developed both of analytical and .... |
.... developed analytical and .... |
|
98 |
in the caption please indicate meaning of dashed lines (absence of crack) |
|
|
121 |
Where z= |
where z= |
|
147 |
Where P |
where P |
|
186 |
... are ignorable ... |
....are negligeable .... |
|
195 |
... micro-crackson .... |
.... micro-cracks on .... |
|
197 |
.... becomes satisfactorily .... |
.... becomes satisfactory .... |
|
225 |
When |
where (authors please note that this is not indented, nor capitalized) |
|
225-272 |
please give derivation or reference for equation 17 |
|
|
231 |
in Figure 3 please correct: The ture |
I believe is: True stress |
|
232 |
PZ of Irwin model |
PZ of Irwin model for mode I |
|
251 |
.... critical length .... |
please explain meaning of ‘critical length’. It is critical length for what ? |
|
259 |
.... thresholded .... |
thresholded ? does not exist; please correct |
|
264 |
is it possible to include a0 in this table? |
|
|
268 |
.... for a stress ratio .... |
... for stress ratio .... |
|
284 |
..... radically distributed .... |
please correct, does not make sense! |
|
292 |
.... soft inclusion .... |
please give some comments concerning inclusions. In the manuscript this is the first time that concept appears! |
|
301-302 |
please include a figure showing specimen and loading |
|
|
303 |
.... flutuated .... |
.... flutuating .... |
|
313 |
.... please give some comments concerning coalescence. Do you use Swift criterion? or? |
|
|
328 |
‘radically’ does not make sense |
|
|
329 |
conclusion #4 is unreadable; please re-write |
|
|
343 |
4th version |
is this 4th ed? if so, please correct |
|
403 |
basic engineering |
Basic Engineering |
Comments on the Quality of English Language
‘Digital Twins to Predict Crack Propagation of Engineering Materials under Different Loads for Sustainable Products’
manuscript Machines (MDPI): machines-2840157-peer-review-v1
by Xu Li, et al.
The title should precisely reflect the content of the paper; the present title is confusing:
1 - The focus of the paper is to solve a fracture mechanics problem: use of LEFM to model FCG of a large crack at the tip of which there are several small cracks with different orientations. Of course fatigue is related to sustainability, but the use of the word sustainability in the title of this paper misleads the reader since it does not reflect the main paper contents.
2 - Equal concern regarding reference to digital twins. The ‘digital twin’ paradigm involves the interplay of numerical / theoretical models and real structures throughout their life-cycle. The present paper concentrates on modelling a fracture mechanics problem, i.e. it may constitute one ‘link’ of the ‘digital twin’; the work is related to digital twins philosophy, but that is certainly not the main emphasis of the paper.
A problem to the paper is the poor quality of use of technical English. This is particularly a problem in the abstract and section 1. They need a full re-writing. The other sections are also full of presentation problems, such as occasional use of words that do not make sense as ‘radically’.
The paper revisits some classical LEFM concepts and tools, as Elasticity based on complex variable following Muskhelishvili and/or Fracture Mechs pioneers as Westergaard. The manuscript includes a large amount of dense equations. The reader would like to know what equations were derived by the present authors, and what other equations are transcribed from the literature. In the last case, the reader wants to have references. These requirements should be taken into account by the present authors in a revision of their manuscript.
References are not presented with a consistent format, and this must be revised/corrected by the authors. More importantly, some references are simply wrong – e.g. refs. 8, 21, whose authors are not properly given/written.
Details:
Examples of main crack with several smaller cracks in the tip region should be provided in the introduction, circa lines 41-42.
Lines 56-57: the use of LEFM with very short cracks is subject to controversy (e.g. concerning length of crack versus plastic region size, grain size, etc.). In lines 56-57 some discussion concerning that controversy, including relevant refs., should be presented.
Lines 67-68: please provide examples and suitable references.
Line 114: case (iv) – why is this case not represented in Figure 2?
Lines 216-217: The authors should not say that it is reasonable to assume that the plastic strain is relatively small. This assumption is implicit in LEFM; therefore they should omit ‘it is reasonable’ and simply state that it is assumed that the plastic region is relatively small.
Typos, etc.:
|
line |
where it is |
should be |
|
13 |
... predict a fatigue life ... |
... predict fatigue life... |
|
13 |
... since any product and structure is made of selected engineering materials. ... |
omit this sentence; it is obvious and does not add anything to the text. |
|
15 |
... is corresponded to the maximum .... |
... is related to the maximum .... |
|
17 |
... initialized angle ... |
.... initial angle ... |
|
18 |
... radically ... |
please correct: ‘radically’ does not make any sense! |
|
41 |
.... damage occurs to single crack ... |
... damage consists of a single crack .... |
|
60 |
Shafique [20] |
Kahn [20] |
|
62 |
Lwa [21] |
Wang [21] |
|
62 |
.... developed both of analytical and .... |
.... developed analytical and .... |
|
98 |
in the caption please indicate meaning of dashed lines (absence of crack) |
|
|
121 |
Where z= |
where z= |
|
147 |
Where P |
where P |
|
186 |
... are ignorable ... |
....are negligeable .... |
|
195 |
... micro-crackson .... |
.... micro-cracks on .... |
|
197 |
.... becomes satisfactorily .... |
.... becomes satisfactory .... |
|
225 |
When |
where (authors please note that this is not indented, nor capitalized) |
|
225-272 |
please give derivation or reference for equation 17 |
|
|
231 |
in Figure 3 please correct: The ture |
I believe is: True stress |
|
232 |
PZ of Irwin model |
PZ of Irwin model for mode I |
|
251 |
.... critical length .... |
please explain meaning of ‘critical length’. It is critical length for what ? |
|
259 |
.... thresholded .... |
thresholded ? does not exist; please correct |
|
264 |
is it possible to include a0 in this table? |
|
|
268 |
.... for a stress ratio .... |
... for stress ratio .... |
|
284 |
..... radically distributed .... |
please correct, does not make sense! |
|
292 |
.... soft inclusion .... |
please give some comments concerning inclusions. In the manuscript this is the first time that concept appears! |
|
301-302 |
please include a figure showing specimen and loading |
|
|
303 |
.... flutuated .... |
.... flutuating .... |
|
313 |
.... please give some comments concerning coalescence. Do you use Swift criterion? or? |
|
|
328 |
‘radically’ does not make sense |
|
|
329 |
conclusion #4 is unreadable; please re-write |
|
|
343 |
4th version |
is this 4th ed? if so, please correct |
|
403 |
basic engineering |
Basic Engineering |
Author Response
Please see in the attached file.

Reviewer 2 Report
Comments and Suggestions for Authors
Dear Authors,
This paper addresses the development of a computer-aided engineering model for a digital twin, aiming to predict the fatigue life of materials under various loads for the promotion of product sustainability. The theme is relevant in the current context of engineering and sustainable development, well-formulated, and supported in appropriate technical language, although certain sections may require increased clarity for broader accessibility.
Recommendations for Improvement:
Introduction:
- Emphasize more clearly the novelty, originality, and necessity of the study to captivate readers' attention from the opening paragraphs.
- Highlight the level of originality of the subject by comparing it with other published materials and explaining the distinct contribution it makes to the field.
Experiment:
- Expand in detail the experimental section to provide a deeper understanding of the methodology used and the experimental conditions.
Results and Discussions:
- Introduce a well-defined section for Results and Discussions, addressing clearly the questions: "What happened?" and "What was discovered or confirmed?" Describe the significance of the data in simple terms.
- Conduct an assessment of observed trends and explain the significance of the results in the context of published research, providing a critical analysis of the collected data.
Conclusions:
- Extend the conclusions section to highlight more extensively the theoretical and practical contribution of the work, offering clear directions for future research.
Bibliography:
- Add at least five recent bibliographic titles (from the last five years) to ensure the relevance and timeliness of references.
Carefully review the manuscript to correct spelling errors, including those in the bibliography, ensuring that the text is coherent and grammatically correct.
In conclusion, congratulations to the authors for the effort put into this innovative work. The recommended improvements aim at clarifying and expanding key sections, ensuring that the theoretical and practical contributions are more evident, providing readers with a broader perspective on the obtained results. These enhancements will strengthen the value and impact of the work in the research field.
Comments on the Quality of English Language
Minor corrections.
Author Response
Please see in the attached file.

Round 2
Reviewer 1 Report
Comments and Suggestions for Authors'Digital Twins to Predict Crack Propagation of Sustainable Engineering Materials under Different Loads', by Xu Li et al.
I still think that the title of the work could be more focused on fracture mechanics (without reference to digital twins and sustainability), but I have no problem in accepting the title chosen by the authors.
The authors revised and improved their manuscript, and the present version is suitable for publication.